# Restoration of Sarcoplasmic Reticulum Ca^2+^ ATPase (SERCA) Activity Prevents Age-Related Muscle Atrophy and Weakness in Mice

**DOI:** 10.3390/ijms22010037

**Published:** 2020-12-22

**Authors:** Rizwan Qaisar, Gavin Pharaoh, Shylesh Bhaskaran, Hongyang Xu, Rojina Ranjit, Jan Bian, Bumsoo Ahn, Constantin Georgescu, Jonathan D. Wren, Holly Van Remmen

**Affiliations:** 1Aging and Metabolism Research Program, Oklahoma Medical Research Foundation, Oklahoma City, OK 73104, USA; rqaisar@sharjah.ac.ae (R.Q.); g.a.pharaoh@gmail.com (G.P.); Shylesh-Bhaskaran@omrf.org (S.B.); Yang-Xu@omrf.org (H.X.); Rojina-Ranjit@omrf.org (R.R.); justbianjan@gmail.com (J.B.); Bumsoo-Ahn@omrf.org (B.A.); 2Department of Basic Medical Sciences, College of Medicine, University of Sharjah, Sharjah 27272, UAE; 3Division of Genomics and Data Sciences, Oklahoma Medical Research Foundation, Oklahoma City, OK 73104, USA; Constantin-Georgescu@omrf.org (C.G.); Jonathan-Wren@omrf.org (J.D.W.); 4Oklahoma City VA Medical Center, Oklahoma City, OK 73104, USA

**Keywords:** skeletal muscle, sarcoplasmic reticulum calcium-transporting ATPase, sarcopenia

## Abstract

Sarcopenia has a significant negative impact on healthspan in the elderly and effective pharmacologic interventions remain elusive. We have previously demonstrated that sarcopenia is associated with reduced activity of the sarcoplasmic reticulum Ca^2+^ ATPase (SERCA) pump. We asked whether restoring SERCA activity using pharmacologic activation in aging mice could mitigate the sarcopenia phenotype. We treated 16-month male C57BL/6J mice with vehicle or CDN1163, an allosteric SERCA activator, for 10 months. At 26 months, maximal SERCA activity was reduced 41% in gastrocnemius muscle in vehicle-treated mice but maintained in old CDN1163 treated mice. Reductions in gastrocnemius mass (9%) and in vitro specific force generation in extensor digitorum longus muscle (11%) in 26 versus 16-month-old wild-type mice were also reversed by CDN1163. CDN1163 administered by intra-peritoneal injection also prevented the increase in mitochondrial ROS production in gastrocnemius muscles of aged mice. Transcriptomic analysis revealed that these effects are at least in part mediated by enhanced cellular energetics by activation of PGC1-α, UCP1, HSF1, and APMK and increased regenerative capacity by suppression of MEF2C and p38 MAPK signaling. Together, these exciting findings are the first to support that pharmacological targeting of SERCA can be an effective therapy to counter age-related muscle dysfunction.

## 1. Introduction

Age-related loss of muscle mass and strength or sarcopenia is a major cause of functional dependency in the elderly [1]. The pathophysiology of sarcopenia is multifactorial [2], with changes at cellular and subcellular levels contributing to muscle detriment. We have previously shown that Ca^2+^ regulation is altered in aging muscle [3]. In particular, the activity of sarcoplasmic reticulum (SR) Ca^2+^ ATPase (SERCA), a key enzyme involved in intracellular Ca^2+^ regulation, is reduced in aging [3] and in pathological conditions mimicking aging [4], which is partly due to increased oxidation [5] and nitrosylation [6] of SERCA. The reduced calcium uptake by the SR leads to an increased accumulation of Ca^2+^ in the cytosol [7], which has pathological consequences. For example, higher cytosolic Ca^2+^ reduces the sensitivity of the contractile apparatus to Ca^2+^, leading to muscle weakness [8]. Ca^2+^ overload also disrupts the mitochondrial function and activates several catabolic processes, further contributing to age-related muscle loss and strength [9]. The resultant oxidative stress and cellular pathologies exacerbate the SR dysfunction by reducing the SR Ca^2+^ transients [7]. Additionally, SERCA dysfunction is not an isolated event and causes the dysregulation of cellular excitation-contraction (EC) coupling machinery. For example, we and others have shown that reduced SERCA activity in aging is associated with degradation and/or reduction of SR Ca^2+^ release channels, namely ryanodine receptors (RyR) and calstabin [3,7]. These changes result in ‘’leaky’’ RyRs with a higher probability of channel opening and leakage of Ca^2+^ ions into the sarcoplasm, leading to reduced SR Ca^2+^ reservoirs and muscle weakness [10]. Reduced SERCA activity also contributes to lower SR Ca^2+^ stores, which compromises the amount of Ca^2+^ released through RyR channels for contraction during muscle activation [11]. Accordingly, an age-related reduction in the SR Ca^2+^ stores has been reported [11]. These findings highlight the pivotal role of SR in the sarcopenia process and suggest that restoration of SERCA may be a promising intervention to counter muscle pathology in aging. In support of this, overexpression of SERCA protein or deletion of the SERCA inhibitory protein, sarcolipin, enhance SERCA enzyme activity, which restores muscle phenotype in a mouse model of Duchenne muscular dystrophy [12,13]. Specifically, fiber regeneration is improved, and cytosolic Ca^2+^ levels, muscle injury, and necrosis are reduced. Despite these promising findings, pharmacotherapy to restore SERCA activity in aging has not been explored.

To date, only a handful of pharmacological activators of SERCA are available [14], and their effects on skeletal muscle mass and strength remain largely unknown. CDN1163 has recently emerged as a potent allosteric activator of SERCA and has shown promising therapeutic effects in Parkinson’s, Alzheimer’s, and metabolic diseases [15,16,17]. We have recently shown that CDN1163 also restores SERCA activity in a mouse model of oxidative stress (*Sod1*^−/−^ mice) with promising therapeutic results, including reduced mitochondrial dysfunction, prevention of muscle atrophy and weakness, and reduced oxidative stress [4]. These findings suggest CDN1163 as an attractive intervention to counter muscle detriment in sarcopenia. Here, we have extended our findings to show that CDN1163 also mitigates the sarcopenia phenotype in aged wild-type mice. C57BL/6J wild-type mice were supplemented with CDN1163 in diet or via intraperitoneal injections, and the effects on SERCA activity, muscle mass and strength and mitochondrial function were investigated.

## 2. Results

### 2.1. CDN1163 Prevents Decline in SERCA ATPase Activity in the Gastrocnemius Muscle of Aging Mice

The activity of the SERCA ATPase enzyme was determined in the lysates from gastrocnemius muscles by measuring the decrease in absorbance of NADH at 340 nm. Aged-control mice showed significantly reduced maximum SERCA ATPase activity (41%, *p* < 0.05) when compared to 16-month old control mice (Figure 1). 8–10 months of treatment with CDN1163 in diet and via intraperitoneal (i.p.) administration completely reversed the loss of SERCA activity in the older mice.

### 2.2. CDN1163 Prevents Muscles Atrophy in the Aging Mice

The aged mice showed significantly reduced body weight irrespective of their treatment status when compared to 16-month old control mice (Figure 2A). The absolute mass of gastrocnemius muscles was also reduced in aged mice in all groups when compared to 16-month old control mice (Figure 2B). However, when the muscle mass was normalized to body weight, only the aged-control mice showed significant atrophy of the gastrocnemius (9%, *p* < 0.05), when compared to 16-month-old mice (Figure 2C) supporting a positive effect of the CDN treatment on gastrocnemius mass in old CDN treated mice. Treatment with CDN1163 supplemented diet completely prevented the atrophy of both gastrocnemius and quadriceps muscles (Figure 2C,D) in aged mice when normalized to body mass, while CDN1163 administered by i.p. injection prevented the atrophy of gastrocnemius muscle only.

### 2.3. CDN1163 Prevents Contractile Dysfunction of the Extensor Digitorum Longus (EDL) Muscle in the Aging Mice

In agreement with our previous findings [3], we measured an age-related reduction in absolute contractile force in the EDL muscles from aged mice without CDN1163 treatment (21%, *p* < 0.05) when compared to 16-month-old mice (Figure 3A). Remarkably, this age-related muscle weakness was completely prevented with CDN1163 supplementation in diet or via intraperitoneal injection. Likewise, when muscle force was normalized to muscle size (specific force), specific force was reduced by 11% in the aged control mice and reversed by CDN1163 (Figure 3B).

### 2.4. CDN1163 Attenuates Mitochondrial Dysfunction in Sarcopenia

There is a tight structural and functional relationship between muscle sarcoplasmic reticulum (SR) and mitochondria [18]. We have previously shown that CDN1163 attenuates mitochondrial dysfunction in a mouse model of increased oxidative stress and accelerated muscle atrophy and weakness (*Sod1*^−/−^ mice) [19]. It is well established that intracellular Ca^2+^ dysregulation and mitochondrial dysfunction have a substantial contribution to sarcopenia [3]. Due to the structural and functional communication between muscle SR and mitochondria [18], we hypothesized that CDN1163-induced restoration of SERCA ATPase might reduce the elevated mitochondrial reactive oxygen species (ROS) production that occurs in aging skeletal muscle. We measured production of mitochondrial hydroperoxides using the Amplex Red probe. In agreement with our previous findings [20] isolated mitochondria from aged control mice showed significantly higher production of hydroperoxides than mitochondria from 16-month-old control mice in State-1 and glutamate male supported respiration (Figure 4A,B). Consistent with these findings, CDN1163 treatment by injection significantly reduced glutamate malate-supported hydroperoxide production in older mice to levels comparable to 16-month-old mice (Figure 4A,B). However, CDN1163 administered in the diet increased mitochondrial hydroperoxide production. The reason for the increased mitochondrial peroxide generation in response to dietary supplementation of CDN1163 is unclear.

### 2.5. CDN1163 Treatment via Diet and Injection Prevents Age-Related Alterations in HSF1, P38 MAPK, MEF2C, PRKAA2 and APMK Pathways

We used RNA seq and Ingenuity upstream regulator analysis to investigate the effects of CDN1163 and age on various pathways controlling skeletal muscle adaptation. Heat shock factor-1 (HSF-1), Myocyte-specific enhancer factor 2C (MEF2C), p-38 mitogen-activated protein kinases (p38 MAPK), 5″-AMP-activated protein kinase catalytic subunit alpha-2 (PRKAA2) and5′AMP-activated protein kinase (AMPK) pathways were identified as potential mediators of muscle restoration in CDN1163 treated mice. More specifically, age was associated with inhibition of HSF-1, PRKAA2, and AMPK and activation of p38 MAPK and MEF2C pathways. CDN1163 injections reversed these changes by activating HSF-1, PRKAA2, and APMK while inhibiting p38 MAPK and MEF2C pathways (Figure 5).

### 2.6. CDN1163 Reverses Selected Changes in Skeletal Muscle Gene Signature with A Wging

Next, we analyzed the gene signature from gastrocnemius muscles in an unbiased manner for changes with age and CDN1163 treatment. Aging resulted in a change in the expression of 871 genes (upregulated = 522; downregulated = 349) in the aged control group when compared to 16-month old adult mice. On the other hand, CDN1163 treatment (combined diet and injection) changed the expression of 317 genes (upregulated = 119; downregulated = 198) when compared to vehicle-treated (combined diet and injection) aged mice. Gene Ontology analysis revealed that sarcomeric and Z-disk cellular components were upregulated by CDN1163 treatment in old mice, while troponin complex cellular components were downregulated (Figure 6). Interestingly, *Atp2a2* (the cardiac muscle, slow twitch SERCA 2) was downregulated by CDN1163 treatment, while *Casr* (Calcium-sensing receptor) was upregulated (Figure 6). Aging also results in a change in fiber-type composition from fast- to slow-twitch fibers, which is partly due to preferential atrophy of fast-twitch fibers [21]. Thus, the aging muscle has a high proportion of slow-twitch fibers, which are resistant to mechanical damage. This transformation of muscle fiber-types is associated with alterations in the expression of several genes encoding sarcomeric myosin and troponin as well as their associated proteins [22]. CDN1163 treatment changes the expressions of several myosin and troponin related genes, which further support the shift towards slow-twitch program in the aging skeletal muscle. This finding is supported by a shift towards slow-twitch fibers in the mice with enhanced SERCA activity due to deletion of sarcolipin [23]. A list of the top 25 up- and down-regulated genes is summarized in Figure 6.

## 3. Discussion

This is the first study showing that direct pharmacological activation of SERCA ATPase can mitigate age-related muscle loss and weakness. The results from this study confirm our previous findings that SERCA activation can be a powerful tool to boost muscle mass in conditions of increased oxidative stress [4]. Here, we extend these findings to include sarcopenia. We also present transcriptomic data pointing to potential molecular mediators of muscle restoration with CDN1163 treatment, including AMPK, p38 MAPK, UCP1, and HSF1 that may play critical roles in SERCA-mediated muscle restoration with CDN1163 treatment.

A key function of the SERCA pump is clearing cytosolic Ca^2+^ after muscle contraction. This critical role of SERCA is essential for cellular maintenance as SERCA dysfunction can lead to an increase in the cytosolic Ca^2+^ levels, which can trigger muscle weakness and atrophy through multiple mechanisms including changes in transcriptional programs [24]. Reduced SERCA activity and increased cytosolic Ca^2+^ also have catabolic consequences, leading to muscle atrophy and weakness [13]. Indeed, compromised function of SERCA has been associated with many muscle diseases [12,25] including aging [3]. Thus, the activation of the SERCA pump can be a powerful approach to prevent muscle decline. Studies to determine the molecular basis of this protection are ongoing in our laboratory.

We have previously shown that aging and oxidative stress are not associated with changes in SERCA protein expression in gastrocnemius muscle [3,4] despite the fact that SERCA activity is reduced. SERCA has a long half-life of 14–17 days, which makes it potentially susceptible to oxidative damage [26]. Accordingly, SERCA dysfunction in aged muscle could be linked at least in part to increased oxidative stress in skeletal muscle. High levels of oxidative stress in skeletal muscle have also previously been linked to increased cytosolic Ca^2+^ and mitochondrial dysfunction [27] which contribute to muscle atrophy and weakness. A direct linkage between SERCA dysfunction and higher cytosolic Ca^2+^ accumulation has been suggested as the restoration of SERCA activity by SERCA overexpression prevents the pathologic increase in cytosolic Ca^2+^ levels [28]. High cytosolic Ca^2+^ also causes mitochondrial Ca^2+^ overload and increased production of mitochondrial ROS. Specifically, ROS produced in the basal state (state 1) in muscle mitochondria following denervation are mostly lipid hydroperoxides, which are mainly produced by calcium-dependent enzyme phospholipase-2 (cPLA_2_) [29]. Thus, we hypothesized that one possible effect of CDN1163-mediated SERCA restoration could be to reduce cytosolic free Ca^2+^ levels and mitochondrial peroxide production. Reduced mitochondrial peroxide generation could, in turn, lower oxidative stress and oxidative stress-induced changes in muscle atrophy and function in agreement with our previous report of SERCA restoration and reduced mitochondrial dysfunction with CDN1163 treatment in the *Sod1^-/-^* mice [4]. There are various potential mechanisms by which SERCA restoration can mitigate mitochondrial dysfunction. Reduced SERCA activity leads to increased cytosolic Ca^2+^ which has catabolic consequences, leading to muscle atrophy and weakness [13]. Moreover, increased cytosolic Ca^2+^ also leads to mitochondrial Ca^2+^ overload and compromised function, which leads to increased ROS production in various muscle pathologies [30]. Thus, SERCA restoration can reduce Ca^2+^ buildup in cytosol and mitochondria, resulting in reduced oxidative stress [4]. In support of this, pharmacological inhibition of SERCA activity modulates mitochondrial Ca^2+^ uptake channels, eliciting the direct functional communication between SR and mitochondrial Ca^2+^ handling [31]. SERCA dysfunction is also associated with the opening of mitochondrial transitional permeability pores (mPTP), which are implicated in the development and progression of the sarcopenia phenotype [32]. Our findings here are not conclusive. While our measures of hydroperoxides generation are reduced in mice treated with CDN1163 administered using the injection protocol, in contrast, mitochondrial hydroperoxides generation was significantly elevated in response to CDN1163 dietary supplementation. These findings show that the mitochondrial ROS production may not be the prime determinant of cellular oxidative stress, and other mechanisms such as cytosolic antioxidant enzymes may play a varying degree of role in controlling cellular redox environment and associated sarcopenia phenotype. Thus, it is not clear from our results whether mitochondrial generation of peroxides are a component of the protective effect of CDN1163 on muscle mass and force generation.

The SERCA pump is a major regulator of Ca^2+^ during the EC coupling process. This is evident by the disruption of several markers of EC coupling in age-associated SERCA dysfunction [3,7]. We hypothesized that CDN mediated-SERCA restoration at-least partly reverses the pathological changes in the EC coupling machinery. In support of this, we report the reversal of age-related changes in several proteins involved in Ca^2+^ regulation (Figure 6B). Of particular interest is the upregulation of beta-1 subunits of voltage-dependent calcium channels (Cacnb1), which encode the DHPR beta-1 receptors. The reduction in DHPR expression has pathological consequences on muscle innervation as well as structural and functional health [33]. Thus, CDN1163 can potentially mitigate these negative changes in skeletal muscle by preventing age-related reduction in this Ca^2+^ sensor. 

Interestingly, the transcriptomic data presented here also suggest that mitochondria may be involved. Specifically, we show that CDN1163 treatment resulted in activation of UCP-1 in skeletal muscle which can potentially further contribute to mitochondrial restoration. In support of this, activation of UCP-1 in skeletal muscle is shown to induce proton leak which in muscle, unlike in brown adipose tissues, results in reduced ROS emission by muscle mitochondria [34]. CDN1163 also increased mRNA expression of PGC1-α in skeletal muscle from old mice, potentially inducing mitochondrial biogenesis and providing healthy mitochondria for cellular bioenergetics. These effects of CDN1163 on skeletal muscle are in agreement with its previously reported effects on hepatocytes, where CDN1163 reduced oxidative stress partly by activating PGC1-α and UCP1 [16].

Moreover, we provide evidence that CDN1163-mediated mitochondrial restoration in skeletal muscle may be driven by activation of AMPK and its catalytic subdomain PRKAA2. Similar findings of APMK activation with CDN1163 treatment have been reported in the liver [16], heart [35], and pancreatic tissues [36]. These findings suggest that SERCA modulation of AMPK signaling and UCP1 may promote mitochondrial biogenesis and reduce ROS emission. In support of these findings, activation of SERCA enzyme activity is shown to reduce mitochondrial ROS production and swelling in skeletal muscles of mice with muscular dystrophies [12]. In addition to its metabolic role, AMPK is also heavily involved in signaling pathways controlling skeletal muscle mass. This is evident by the suppression of AMPK signaling in age-related muscle loss [14] which offers AMPK activation as an attractive molecular target to mitigate muscle wasting in various diseases [37]. Thus, our finding of AMPK activation in CDN1163 treated mice is a potential contributor to maintained muscle mass in aging. These observations suggest that CDN1163 can prevent sarcopenia phenotype, at least partly by the restoration of SERCA and mitochondrial functions.

The transcriptomic data also suggest that HSF1, p38 MAPK, and MEF2C, critical regulators of skeletal muscle mass and regeneration may play a role in the restoration of muscle mass and function in the CDN1163 treated aged mice. HSF1 associated stress response promotes skeletal muscle hypertrophy during increased loading as evident by reduced hypertrophy of soleus after synergistic ablation of hind-limbs plantaris and gastrocnemius muscles in HSF1 deficient mice [38]. Aging is associated with attenuation of HSF1 which is implicated in multiple age-related diseases [39]. On the other hand, activation of p38 MAPK exacerbates skeletal muscle detriment which involves among other factors, attenuation of stem cell self-renewal [40]. We do not know the exact mechanism(s) by which CDN1163 treatment resulted in activation of HSF1 and suppression of p38 MAPK in sarcopenia. However, functional coupling between heat shock proteins and SERCA activity has been described before [41] and can potentially explain CDN1163-mediated restoration of SERCA activity as an activator of HSF1 in sarcopenia. On the other hand, oxidative stress is a major inducer of p38 MAPK [42]. MEF2 family of transcription factors has four isoforms including MEF2a, b, c, and d. Among them, MEF2c plays a critical role in skeletal muscle growth and regeneration during denervation [43]. Thus, the activation of MEF2c during aging may contribute to cellular remodeling in sarcopenia. Suppression of MEF2c by CDN1163 suggests that CDN1163 inhibits denervation-related transcriptional profiling in aging skeletal muscle.

In conclusion, we have shown for the first time that pharmacological activation of SERCA by CDN1163 can be a potent intervention to reduce age-related muscle loss and weakness. We also propose the preservation of mitochondrial function and changes in key signaling pathways as candidate mechanisms for CDN1163-mediated mitigation of the sarcopenia phenotype. Currently, no molecular compound has been shown to preserve muscle mass and strength during aging. Our findings show that CDN1163 can be an effective therapy to offset sarcopenia. It should be noted that while we did not investigate the off-target effects of CDN1163 in this study, this compound has previously been shown to lack off-target effects when screened for 160 potential targets [15,17]. Moreover, genetic activation of SERCA by overexpression of SERCA protein [12] or deletion of sarcolipin [13] show no long-term off-target effects. Future studies will be conducted to validate the efficacy of CDN1163 and to further investigate and validate the mechanistic links between the compound and its positive effects in sarcopenia.

## 4. Material and Methods

### 4.1. Animals and Study Design

Male C57BL/6J wild-type mice were injected intra-peritoneally three times per week starting at 16 months of age for 10 months with vehicle (10% DMSO, 10% Tween-80 in PBS *n* = 12) or CDN1163 (50 mg/kg body weight; *n* = 8). Mice were euthanized at 26 months of age. A control group (*n* = 9) at 16 months was used as the reference point for age comparisons. CDN1163 was purchased from Tocris Bioscience (Minneapolis, MN, USA).

### 4.2. SERCA Assay

SERCA ATPase enzyme activity was measured in muscle homogenates at 37 °C using a spectrophotometric assay as previously described [3,4]. Muscle samples were homogenized 1:10 with SERCA homogenizing buffer, containing (in mM) 250 sucrose, 5 HEPES, 0.2 PMSF, 0.2% NaN_3_. Then mix the homogenate with the protein amount of 100 µg into the SERCA assay buffer containing (in mM) 200 KCl, 20 HEPES, 10 NaN_3_, 1 EGTA, 15 MgCl_2_, 5 ATP, 10 phosphoenolpyruvate, to make a 3 mL mixture, then 18 U/mL of lactate dehydrogenase and pyruvate kinase (PK/LDH), and 1 mM Ca^2+^ ionophore A-23187 (C-7522; Sigma-Aldrich, St. Louis, Missouri, USA) were added into the mixture. These reaction mixtures were then aliquoted and mixed with CaCl_2_ to form 7 pCa points from 7.6 to 4.2 and a blank, and then loaded into a pre-warmed 37 °C quartz plate. The reaction was initiated by adding 1 mM NADH, and the kinetic assay was done (Temperature = 37 °C, Time = 30 min, λ = 340 nm, shaking between readings). The SERCA activity was calculated using the formula:(1)$$\mathrm{Total}\;\mathrm{ATPase}\;\mathrm{rate}\;=\;\frac{{\mathrm{rate}\;\mathrm{of}\;A}_{340\;\mathrm{nm}}\;\mathrm{signal}\;\mathrm{loss}\;}{\mathrm{pathlength}\;\times \;6{.23\;\mathrm{mM}}^{-1}{\mathrm{cm}}^{1}}$$

### 4.3. Muscle Contractile Function

Contractile force generation was measured in the isolated EDL muscles using a 1200A in-vitro test system (Aurora Scientific Inc., Aurora ON, Canada) as described previously [3,4]. Briefly, muscles were individually tied to a model 300C servomotor (Aurora Scientific Inc., Aurora, ON, Canada) and fixed within a water bath containing an oxygenated (95% O_2_, 5% CO_2_) Krebs–Ringer solution (in mM: 137 NaCl, 5 KCl, 1 MgSO_4_, 1 NaH_2_PO_4_, 24 NaHCO_3_, 2 CaCl_2_) maintained at 32 °C. Computer-controlled stimulation was applied through a model 701C stimulator (Aurora Scientific Inc., Aurora, ON, Canada). All data were recorded and analyzed using commercial software (DMC and DMA, Aurora Scientific, Aurora, ON, Canada). Specific force (N/cm^2^) was calculated using an estimated muscle cross-sectional area (CSA) based on muscle mass and length.

### 4.4. Mitochondrial Hydroperoxides Generation

Mitochondrial isolation was performed in gastrocnemius muscle as described previously [44] and freshly isolated mitochondria were used to measure the generation of hydroperoxides using the Amplex red-horseradish peroxidase (HRP) method [20]. HRP (1 U/mL) catalyzes the hydroperoxides-dependent oxidation of nonfluorescent Amplex red (80 µM) (Molecular Probes, Eugene, OR, USA) to the fluorescent compound resorufin. We have previously shown that Amplex Red reacts with hydroperoxides, including both H_2_O_2_ and lipid hydroperoxides, released by isolated mitochondria, and that lipid hydroperoxides are a commonly measured species in muscles under conditions involving loss of innervation [29]. Fluorescence was measured at an excitation wavelength of 544 nm and emission wavelength of 590 nm using a Fluoroskan Ascent type 374 multi-well plate reader (Labsystems, Helsinki, Finland). The slope of the increase in fluorescence is converted to the rate of H_2_O_2_ production with a standard curve produced using increasing H_2_O_2_ concentrations. All assays were performed at 37 °C in 96-well plates. Substrates used were 10 mM succinate plus 1µM rotenone and 5 mM glutamate plus 5mM malate. For each assay, one reaction well-contained buffer only, and another contained buffer with mitochondria, to estimate the background oxidation rates of Amplex Red and to estimate the rate of peroxide release in mitochondria without substrate (state 1). The reaction buffer consisted of 125 mM KCl, 10 mM HEPES, 5 mM MgCl_2_, 2 mM K_2_HPO_4_, and 37.5 U/mL Superoxide dismutase, pH 7.44 [44].

### 4.5. Transcriptome Sequencing Data Generation

Libraries were prepared using the TruSeq Stranded mRNA Library Kit (Illumina, San Diego, CA, USA). The libraries were then sequenced on an Illumina NextSeq 500 (Illumina, San Diego, CA, USA) to produce paired-end 75 base pair reads. The data collected were given to Discovery Bioinformatics Core at the Oklahoma Nathan Shock Center of Excellence in the Biology of Aging for data analysis.

RNA-seq data processing followed the guidelines and practices of the ENCODE and modENCODE consortia regarding proper experimental replication, sequencing depth, data and metadata reporting, and data quality assessment (https://www.encodeproject.org/documents/cede0cbe-d324-4ce7-ace4-f0c3eddf5972/). Raw sequencing reads (in a FASTQ format) were trimmed of residual adaptor sequences using Scythe software (Bioinformatics Core, University of California, Davis, CA, USA). Low-quality bases at the beginning or the end of sequencing reads were removed using sickle then the quality of remaining reads was confirmed with FastQC. Further processing of quality sequencing reads was performed with utilities provided by the Tuxedo Suite software (Oracle Corporation, Redwood City, CA, USA). Reads were aligned to the *Mus musculus* genome reference (GRCm38/mm10) using the TopHat component, then cuffquant and cuffdiff were utilized for gene-level read counting and differentially expression analysis. A false discovery rate threshold of 0.05 was used as selection criteria for differentially expressed genes between pairs of time points. Functional analysis to find overrepresented functional sets (GO, KEGG pathways) was performed using specialized R Bioconductor packages. Ingenuity Pathway Analysis (IPA, QIAGEN, Redwood City CA, USA, https://www.qiagenbioinformatics.com/products/ingenuitypathway-analysis) was used to explore significant gene networks and pathways interactively.

### 4.6. Differential Gene Expression Analysis

Gene expression was quantified by the number of reads mapped to the sense-strand exons and converted to reads per kilobase per million (RPKM). Genes were flagged as detectable with an empirical minimum RPKM of 0.1 and fold changes were computed as the ratio between the arithmetic mean RPKM values of the groups. Genes with a fold change ≥ 1.5 in either direction or with a *p* < 0.05 were reported as significantly differentially expressed genes (DEGs).

### 4.7. Ingenuity Upstream Regulator and Pathway Analysis

The topmost significantly perturbed skeletal muscle genes in the mice treated with CDN1163 when compared to vehicle-treated mice were subjected to ingenuity upstream analysis (IPA, QIAGEN Redwood City, CA, USA, www.qiagen.com/ingenuity). The most significant upstream analysis results were reported based on *p* < 0.0001 and activation *z*-score > 2.

### 4.8. Statistical Analysis

All numerical values are presented as mean ± SEM and the comparisons among the four groups were performed by one-way analysis of variance (ANOVA) and Tukey’s multiple comparison tests, with a single pooled variance. Data were analyzed using GraphPad Prism 9 (GraphPad Software, La Jolla, CA, USA) and * *p* ≤ 0.05, ** *p* ≤ 0.01, *** *p* ≤ 0.001, **** *p* ≤ 0.0001, were considered statistically significant.

## Figures and Tables

**Figure 1 ijms-22-00037-f001:**
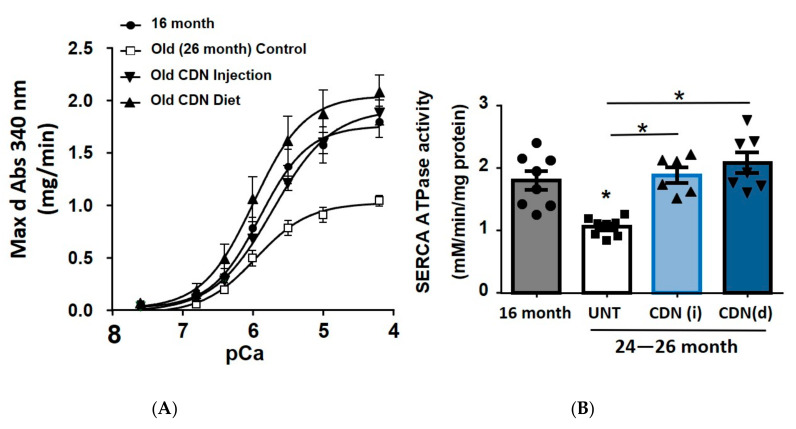
SERCA activity. CDN1163 treatment resulted in restoration of sarcoplasmic reticulum Ca^2+^ ATPase (SERCA) enzyme activity as shown by the SERCA activity—pCa curves (**A**) and maximal SERCA activity (**B**) in the 26 months old, aged mice when compared to vehicle treated control group. Values are expressed as mean ± SEM; (*n* = 8–9 mice/group); one-way analysis of variance. * *p* ≤ 0.05.

**Figure 2 ijms-22-00037-f002:**
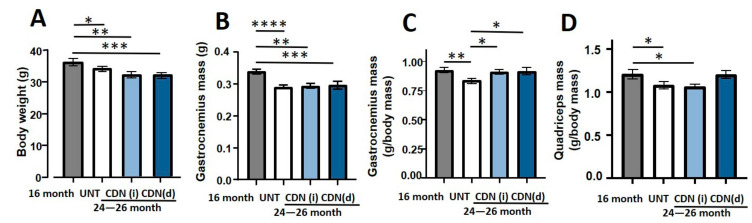
Body weights and muscle masses. CDN1163 treatment did not alter the age-related change in the body weight (**A**) and absolute mass of gastrocnemius muscle (**B**). However, when muscle mass was normalized for body weights, CDN1163 prevented atrophy of gastrocnemius (**C**) and quadriceps muscles (**D**), albeit with the diet only, in the 26 months old, aged mice, when compared to vehicle treated control group. Values are expressed as mean ± SEM; (*n* = 8–9 mice/group); one-way analysis of variance. * *p* ≤ 0.05, ** *p* ≤ 0.01, *** *p* ≤ 0.001, **** *p* ≤ 0.0001.

**Figure 3 ijms-22-00037-f003:**
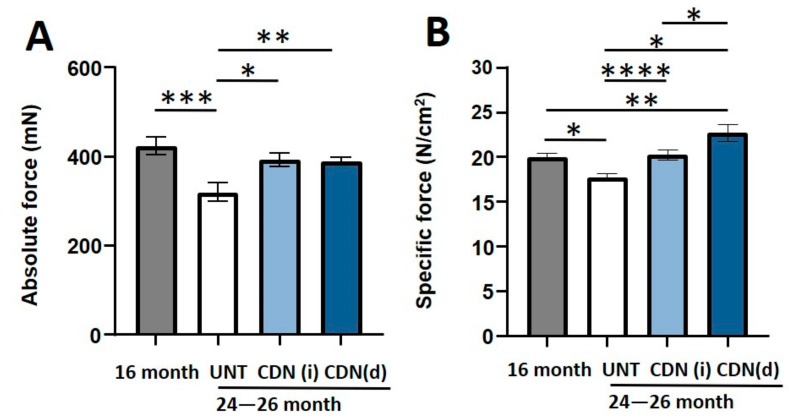
Force generating capacity. CDN1163 prevented the age-related reduction in the absolute (**A**) and specific force (**B**) of extensor digitorum longus (EDL) muscles in the 26 months old, aged mice when compared to vehicle treated control group. Values are expressed as mean ± SEM; (*n* = 8–9 mice/group); one-way analysis of variance. * *p* ≤ 0.05, ** *p* ≤ 0.01, *** *p* ≤ 0.001, **** *p* ≤ 0.0001.

**Figure 4 ijms-22-00037-f004:**
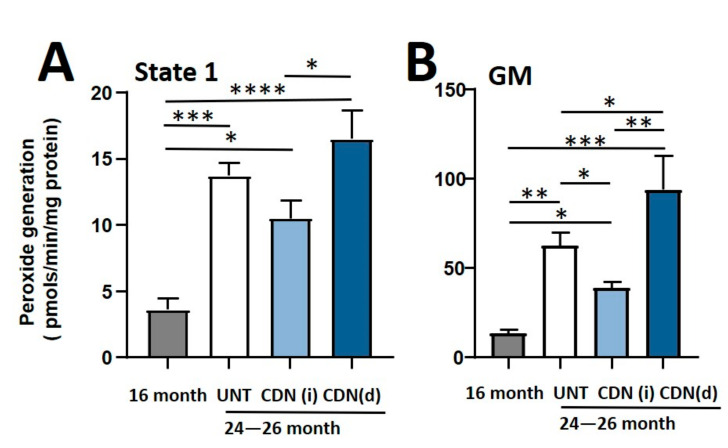
Mitochondrial reactive oxygen species (ROS) production. CDN1163 injections but not diet reduced the mitochondrial ROS production during state-1 (**A**) (without mitochondrial substrate) and in the presence of respiratory substrates (**B**) (glutamate/malate; G/M) in the gastrocnemius muscles of 26-month-old aged mice when compared to vehicle treated control group. Values are expressed as mean ± SEM; (*n* = 8–9 mice/group); one-way analysis of variance. * *p* ≤ 0.05, ** *p* ≤ 0.01, *** *p* ≤ 0.001, **** *p* ≤ 0.0001.

**Figure 5 ijms-22-00037-f005:**
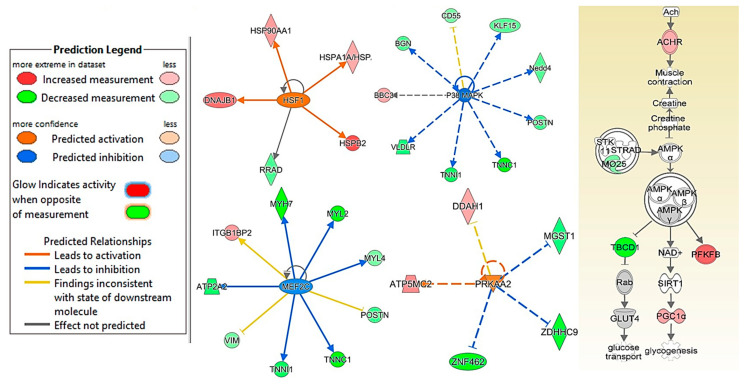
HSF1, p58, MAPK and AMPK signaling. CDN treatment prevented the age-associated alterations in the HSF1, P38 MAPK, MEF2C, PKAA2, and APMK signaling pathways in the gastrocnemius muscles of 26-month old mice, when compared to vehicle treated mice (*n* = 8–9 mice/group). Upstream regulator analysis of HSF1, P38 MAPK, MEF2C, and PKAA2, while pathways analysis of AMPK is shown. The solid lines show the known direct effect, while the dashed lines show known indirect effects.

**Figure 6 ijms-22-00037-f006:**
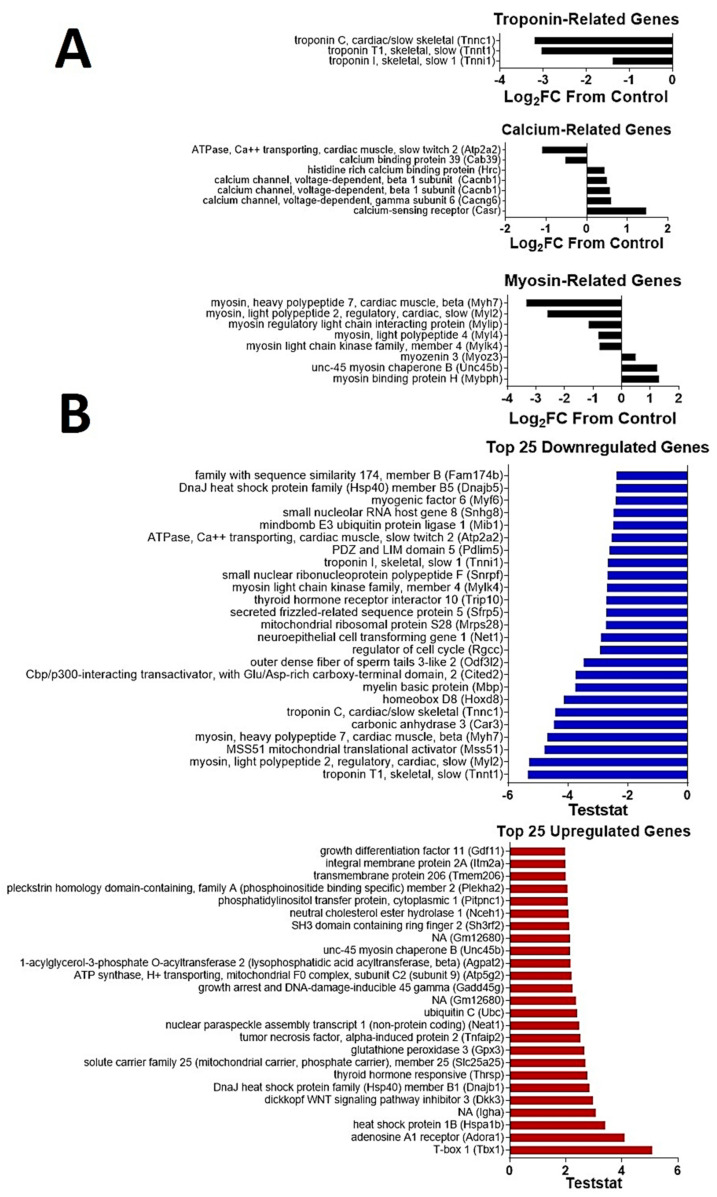
Changes in gene signature with CDN1163 treatment. Ingenuity pathway analysis showing the alterations in the expressions of genes related to troponin, calcium regulation and myosin (**A**) as well as the top 25 up- and down-regulated genes (**B**) in the gastrocnemius muscles of CDN1163-treated 26-month old mice, when compared to vehicle treated mice (*n* = 8–9 mice/group).

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
