# Peer review of "Restoration of Sarcoplasmic Reticulum Ca2+ ATPase (SERCA) Activity Prevents Age-Related Muscle Atrophy and Weakness in Mice"

_ijms, 2020, doi:10.3390/ijms22010037_

Round 1

Reviewer 1 Report

This is an interesting paper on the effects of an allosteric SERCA activator in aged skeletal muscle. The methods used are very clever and the results have important human relevance but there are some issues which should be addressed.

Major points:

The Introduction is quite short. I miss some fresh citation on the effects of aging in skeletal muscle from other research groups. For example Fodor et al. (Sci Rep. 2020;10(1):1707) published a similar animal model to investigate the effects of aging on calcium homeostasis of skeletal muscle. This paper shows SERCA expression and resting cytosolic calcium concentration measurements which is relevant in the present manuscript.

The Authors hypothesized in the Discussion (row 194-196) that “a possible effect of CDN1163-mediated SERCA restoration could be to reduce cytosolic free Ca2+ levels”. Actually it can be proved easily with ratio-metric calcium indicators (like Fura-2) in skeletal muscle fibers. This hypotheses implies more ways to reduce the cytosolic free Ca2+ level. The possible effects of CDN1163 on the key proteins of EC-coupling (RyR, DHPR) and other calcium extrusion ways (sarcolemma and mitochondrial calcium pump) should be considered and discussed.

The second part of the hypothesis also should be approved too, since it is also acknowledged by the authors (row 205) that their result on mitochondrial hydro peroxides generation is controversial (Fig. 4B). This should be explained in the Discussion.

The Authors used appropriate statistical analysis but its results are not correctly presented. For example it is hard to believe that all significant differences reach only the 0.05 level in Fig. 4A and B.

Fig. 5 should be described in more detail. It is not clear why the alterations of AMPK presented on a different way. The size of prediction legend is quite small. Why the Authors used dashed lines in the middle panels?

It is also unclear what the color code means in the tables? Are the values presented in the tables single measurements or averages? In the latter case the errors should be given. Why values are missing from the table for some cells? How the Aged Veh vs CDN Inj. values were calculated and why their majority is missing?

The results presented in Fig. 6 should be discussed in more detail. It is known long time ago that the fiber type composition changes during aging. Examining Fig. 6, the data suggests that CDN1163 treatment shift the fiber type to slow in gastrocnemius muscle. This must be paralleled with the effects of aging.

Minor points:

Abstract:

The abbreviation of extensor digitorum longus is unnecessary. Please omit or define all abbreviation.

Methods:

The gender of animals used should be given.

Some supplier or producer affiliation is missing. See row 312, 316, 320, 322, and 327.

It is stated in row 278 that 8 pCa points were measured however Fig. 1 shows only 7 points.

Results:

The panels in Fig. 1 should be lettered.

I suggest to standardize the graph types. I prefer Fig. 2A and I do not like Fig. 2B and C, because in the latter case the individual points hide the error bars. If the Authors want to show individual data points please use box plot. I have the same remark for Fig. 3.

The unit of Absolute force in Fig. 3A is incorrect, it should be mN.

The sentence should be extended with the correct comparing group in row 154-155.

The legend of Fig. 6 should be extended with “CDN1163 treated” in row 167.

Supplementary figure 1 is not mentioned in the manuscript.

Author Response

Reviewer #1 (Remarks to the Author):

Major points:

Point 1. The Introduction is quite short. I miss some fresh citation on the effects of aging in skeletal muscle from other research groups. For example, Fodor et al. (Sci Rep. 2020;10(1):1707) published a similar animal model to investigate the effects of aging on calcium homeostasis of skeletal muscle. This paper shows SERCA expression and resting cytosolic calcium concentration measurements which is relevant in the present manuscript.

Response 1. We have expanded the introduction section by elaborating on the roles of SR and calcium in sarcopenia process and have also cited the findings of Fodor et al.  on P-1, L-40 to P-2, L-55.

Point 2. The Authors hypothesized in the Discussion (row 194-196) that “a possible effect of CDN1163-mediated SERCA restoration could be to reduce cytosolic free Ca2+ levels”. Actually it can be proved easily with ratio-metric calcium indicators (like Fura-2) in skeletal muscle fibres. This hypotheses implies more ways to reduce the cytosolic free Ca2+ level. The possible effects of CDN1163 on the key proteins of EC-coupling (RyR, DHPR) and other calcium extrusion ways (sarcolemma and mitochondrial calcium pump) should be considered and discussed.

Response 2. The point is well taken. However, we no longer have tissue or treated animals available to perform the ratio-metric assays for measurements of the cytosolic Ca2+ concentrations. We, nevertheless hypothesize that restoration of SERCA activity will reduce the cytosolic Ca2+ levels, as measured by fura-2 ratio-metric assay. A similar finding has been reported in mdx mice where SERCA restoration by genetic overexpression of SERCA protein reduced the resting fura-2 ratio indicating reduced cytosolic Ca2+ (Mazala et al, Cell Physiol, 2015). We have added this information as well as additional information about mitochondrial calcium handling with references in the discussion section, P-8, L-206-209, L-222-226, and P-9L-2350-243.

Point 3. The second part of the hypothesis also should be approved too, since it is also acknowledged by the authors (row 205) that their result on mitochondrial hydro peroxides generation is controversial (Fig. 4B). This should be explained in the Discussion.

Response 3. We have added additional explanation on P-8, L-230-233.

Point 4. The Authors used appropriate statistical analysis but its results are not correctly presented. For example it is hard to believe that all significant differences reach only the 0.05 level in Fig. 4A and B.

Response 4. We have re-run the statistical analysis and have revised the figures 1, 2, 3 & 4 with added numbers of asterisks according to the strength of p value.

Point 5. Fig. 5 should be described in more detail. It is not clear why the alterations of AMPK presented on a different way. The size of prediction legend is quite small. Why the Authors used dashed lines in the middle panels?.

Response 5. The figure shows pathway analysis for the AMPK pathway and the upstream regulator analysis for other pathways. The solid lines show the known direct effects while dashed lines show the known indirect effects. We have added this information to the figure legend and have also increased the size of prediction legend.

Point 6. It is also unclear what the colour code means in the tables? Are the values presented in the tables single measurements or averages? In the latter case the errors should be given. Why values are missing from the table for some cells? How the Aged Veh vs CDN Inj. values were calculated and why their majority is missing?

Response 6. The Tables were erroneously carried over from a previous version of the manuscript and have now been removed.  We apologize for this oversight.

Point 7. The results presented in Fig. 6 should be discussed in more detail. It is known long time ago that the fiber type composition changes during aging. Examining Fig. 6, the data suggests that CDN1163 treatment shift the fiber type to slow in gastrocnemius muscle. This must be paralleled with the effects of aging.

Response 7. The point is well taken, and we have added additional information about the potential shift in muscle fibre-type composition with CDN1163 treatment on P-7, L-168-176.

Minor points:

Point 8. Abstract:

The abbreviation of extensor digitorum longus is unnecessary. Please omit or define all abbreviation.

Response 8. We have omitted the abbreviation.

Point 9. Methods:

The gender of animals used should be given.

Response 9. The sex of animals is given on P-10, L-297.

Point 10. Some supplier or producer affiliation is missing. See row 312, 316, 320, 322, and 327.

Response 10. We have added the supplier’s information.

Point 11. It is stated in row 278 that 8 pCa points were measured however Fig. 1 shows only 7 points.

Response 11. We have corrected the text on P-10, L-312 in the revised manuscript.

Point 12. Results:

The panels in Fig. 1 should be lettered.

Response 12. We have done that.

Point 13. I suggest to standardize the graph types. I prefer Fig. 2A and I do not like Fig. 2B and C, because in the latter case the individual points hide the error bars. If the Authors want to show individual data points please use box plot. I have the same remark for Fig. 3.

Response 13. The graphs in Figures 1, 2 and 3 have been standardized and individual data points removed.

Point 14. The unit of Absolute force in Fig. 3A is incorrect, it should be mN.

Response 14. We have corrected that.

Point 15. The sentence should be extended with the correct comparing group in row 154-155.

Response 15. We have corrected the sentence on P-7, L-161-162.

Point 16. The legend of Fig. 6 should be extended with “CDN1163 treated” in row 167.

Response 16. We have corrected the legend of figure 6 on P-7, L-181.

Point 17. Supplementary figure 1 is not mentioned in the manuscript.

Response 17. We have added the supplementary figure to Figure 2D and mentioned it in the text on P-3, L-92.

Reviewer 2 Report

The paper refers to an important topic, the relation of SERCA activity to muscle contraction in physiology and in pathology. Results on changes of SERCA activity are sound and coherent. Gene expression analysis found activation of PGC1-ꭤ, UCP1, HSF1 and APMK, and suppression of MEF2C and p38 MAPK signaling. The essentials seem clear and well documented, and the SERCA activator CDN1163 rescued from aging sarcopenia. Results are clear and conclusion well founded.

Contribution of oxidative damage and mitochondria is also tested and discussed. Most of the data support mitochondrial involvement, Resting [Ca2+] inside mitochondria is usually assumed to be very low (Spät et al, 2012; Cell Calcium 52:64-72.

The weakest is the biophysical flank. From the point of view of changes of [Ca2+]ER, biophysics predicts that when SERCA activity declines then the pump/leak steady state to decreased of [Ca2+]ER, and this would conduce to smaller ER calcium release during activation and decrease of [Ca2+]C peak and decrease of muscle tension (Delrio-Lorenzo et al. 2020 J Cell Sci. 133(6):jcs240879.) Increase of ER leak through RyR (or other means) would conduce also to decrease of [Ca2+]ER and muscle strength (Andersson et al. 2011 Cell Metabolism, 14: 196-207). Please introduce these references and brief comments in introduction and/or discussion.

In bar diagrams in Figs, 1-3 it would be clearer if individual data (circles) were omitted. SEM already informs us on dispersion and it is more than enough. Do as in fig. 4.

Results on gene expression would perhaps deserve a bit larger comment in the Abstract

Author Response

Reviewer # 2 (Remarks to the Author):

Point 1. The paper refers to an important topic, the relation of SERCA activity to muscle contraction in physiology and in pathology. Results on changes of SERCA activity are sound and coherent. Gene expression analysis found activation of PGC1-ꭤ, UCP1, HSF1 and APMK, and suppression of MEF2C and p38 MAPK signalling. The essentials seem clear and well documented, and the SERCA activator CDN1163 rescued from aging sarcopenia. Results are clear and conclusion well founded. Contribution of oxidative damage and mitochondria is also tested and discussed. Most of the data support mitochondrial involvement, Resting [Ca2+] inside mitochondria is usually assumed to be very low (Spät et al, 2012; Cell Calcium 52:64-72.

Response 1. We are thankful for the positive comments.

Point 2. The weakest is the biophysical flank. From the point of view of changes of [Ca2+]ER, biophysics predicts that when SERCA activity declines then the pump/leak steady state to decreased of [Ca2+]ER, and this would conduce to smaller ER calcium release during activation and decrease of [Ca2+]C peak and decrease of muscle tension (Delrio-Lorenzo et al. 2020 J Cell Sci. 133(6):jcs240879.) Increase of ER leak through RyR (or other means) would conduce also to decrease of [Ca2+]ER and muscle strength (Andersson et al. 2011 Cell Metabolism14: 196-207). Please introduce these references and brief comments in introduction and/or discussion.

Response 2. The point is well taken, and we have added brief comments about SR calcium stores and leak in the introduction section on P-1, L-40 to P-2, L-55.

Point 3. In bar diagrams in Figs, 1-3 it would be clearer if individual data (circles) were omitted. SEM already informs us on dispersion and it is more than enough. Do as in fig. 4.

Response 3. These figures have been revised.

Point 4. Results on gene expression would perhaps deserve a bit larger comment in the Abstract.

Response 4. We have expanded the comment on transcript analysis in the abstract.

We appreciate the very helpful suggestions of the reviewers and the editor and we hope our submission is now acceptable for publication in the International Journal of Molecular Science.

Round 2

Reviewer 1 Report

The Manuscript is improved. The Authors answered all my concerns.

Minor points:

I would not use the abbreviation EDL in the abstract. Please use the whole name (extensor digitorum longus) here.

The Authors incorporated Supplementary figure 1 into Figure 2, but the legend of the figures was not updated. Please mention panel D here.

Author Response

Reviewer #1 (Remarks to the Author):

The Manuscript is improved. The Authors answered all my concerns.

Minor points:

Point 1. I would not use the abbreviation EDL in the abstract. Please use the whole name (extensor digitorum longus) here.

Response 1. We have now used the full name now in the abstract.

Point 2. The Authors incorporated Supplementary figure 1 into Figure 2, but the legend of the figures was not updated. Please mention panel D here.

Response 2. We have updated the legend of figure 2 now.

We appreciate the very helpful suggestions of the reviewers and the editor and we hope our submission is now acceptable for publication in the International Journal of Molecular Science.